# Molecular Epidemiology and Whole-Genome Analysis of Bovine Foamy Virus in Japan

**DOI:** 10.3390/v13061017

**Published:** 2021-05-28

**Authors:** Hirohisa Mekata, Tomohiro Okagawa, Satoru Konnai, Takayuki Miyazawa

**Affiliations:** 1Center for Animal Disease Control, University of Miyazaki, Miyazaki 889-2192, Japan; 2Department of Advanced Pharmaceutics, Faculty of Veterinary Medicine, Hokkaido University, Sapporo 060-0818, Japan; okagawa@vetmed.hokudai.ac.jp (T.O.); konnai@vetmed.hokudai.ac.jp (S.K.); 3Department of Disease Control, Faculty of Veterinary Medicine, Hokkaido University, Sapporo 060-0818, Japan; 4Laboratory of Virus-Host Coevolution, Institute for Frontier Life and Medical Sciences, Kyoto University, Kyoto 606-8507, Japan; takavet@infront.kyoto-u.ac.jp

**Keywords:** genotype, Japan, phylogeny, prevalence, spumavirus

## Abstract

Bovine foamy virus (BFV) is a member of the foamy virus family in cattle. Information on the epidemiology, transmission routes, and whole-genome sequences of BFV is still limited. To understand the characteristics of BFV, this study included a molecular survey in Japan and the determination of the whole-genome sequences of 30 BFV isolates. A total of 30 (3.4%, 30/884) cattle were infected with BFV according to PCR analysis. Cattle less than 48 months old were scarcely infected with this virus, and older animals had a significantly higher rate of infection. To reveal the possibility of vertical transmission, we additionally surveyed 77 pairs of dams and 3-month-old calves in a farm already confirmed to have BFV. We confirmed that one of the calves born from a dam with BFV was infected. Phylogenetic analyses revealed that a novel genotype was spread in Japan. In conclusion, the prevalence of BFV in Japan is relatively low and three genotypes, including a novel genotype, are spread in Japan.

## 1. Introduction

Bovine foamy virus (BFV), formerly called bovine syncytial virus, bovine spumavirus and bovine spumaretrovirus, is a member of the foamy virus (FV) sub-family that includes simian, prosimian, feline, and equine FVs [1]. FVs belong to the sub-family *Spumaretrovirinae* in the family *Retroviridae*, and BFV belongs to the genus *Bovispumavirus*. The BFV genome consists of approximately 12,000 nucleotides, including three structural genes (*gag*, *pol*, *env*) common in retroviruses and two accessory genes (*bel-1*, *bel-2*) characteristic of FVs, which are flanked by two identical long terminal repeats (LTRs). The *bel-1* gene encodes the Tas protein, which functions as a transcriptional activator [2]. The *bel-2* gene may encode the Bel-2 protein, but such a protein has never been unambiguously detected in any FV analyzed so far [3]. Conversely, spliced mRNA from the early portion of the sequence of *bel-1* and the entire sequence of *bel-2* encodes the Bet protein [4]. Bet proteins of simian and feline FVs function as antagonists of antiretroviral proteins (APOBEC3) [5,6]. However, the function of the Bet protein of BFV is still obscure. BFV expresses microRNAs (miRNAs) in vitro and in vivo, and one of them is known to support virus replication [7,8]. BFV is considered a nonpathogenic retrovirus, as are other FVs [9,10]. The oral mucosa is the most active replication site of simian and feline FVs [11,12], and close contact is the essential factor of BFV transmission [13]. Thus, the major route of BFV transmission is considered to be licking or sneezing by BFV-infected cattle. Perinatal transmission through colostrum or milk has been proposed since virus isolation [14]. However, the importance and frequency of perinatal transmission under natural conditions are still controversial.

Although the number of reports is limited, BFV is thought to be endemic throughout the world [14,15,16]. The positive rates have been shown to be markedly different between countries and BFV’s prevalence has ranged from 7% to 42% since 2000 [14,15]. Recently, BFV was first isolated in Japan [17], and two epidemiological surveys were reported [18,19]. These studies reported that the virus was endemic in the Kanto, Hokkaido, and Kyushu areas of Japan, and the prevalence ranged from 12% to 16%. The surveys were mostly conducted in the Kanto area, and one farm each was sampled in the Hokkaido, Tohoku, and Kyushu areas. Thus, an additional survey somewhere other than the Kanto area is required to understand BFV infection in Japan (Figure 1).

FVs are thought to be ancient retroviruses and to have coevolved with host species [20,21]. Due to their nonpathogenic nature, broad tissue tropism, high packaging capacity and low genotoxicity, FVs are under development for novel viral vectors for gene delivery and vaccination [22,23,24]. Furthermore, a recent study suggested that FVs might have an antitumor effect through miRNA expression [25]. Thus, whole-genome analyses of FVs are important for developing viral vectors and for understanding evolutionary biology.

In this study, we analyzed the prevalence of BFV in the Kyushu and Hokkaido areas in Japan, the possibility of vertical transmission and the whole viral sequences of 30 BFV isolates. This study will contribute to the understanding of the character and evolution of BFV.

## 2. Materials and Methods

### 2.1. Samples

Blood samples used for a molecular epidemiological survey were collected from a total of 884 cattle from 89 farms in the Hokkaido (12 farms: *n* = 100), Kumamoto (14 farms: *n* = 140), Oita (1 farm: *n* = 24), Miyazaki (60 farms: *n* = 600), Kagoshima (1 farm: *n* = 10) and Okinawa (1 farm: *n* = 10) prefectures in Japan from November 2016 to March 2020 by local clinical veterinarians. The number of samples per farm was 10 except for a farm in Oita (*n* = 24) and 4 farms (*n* = 6, 6, 5, and 3) in the Hokkaido prefecture. The samples were collected from cattle with a broad age range (0–48 months old: *n* = 255, 48–84 months old: *n* = 265, 84–120 months old: *n* = 221, >120 months old: *n* = 110, unknown: *n* = 33) and mainly from Japanese Black cattle (Holstein: *n* = 116, Japanese Black: *n* = 757, cross-breed: *n* = 9, others: *n* = 2). Apart from the above, additional samples were collected from 77 Holstein pairs of dams and 3-month-old calves on a farm with confirmed BFV infection in June 2020 to assess the possibility of dam-to-calf transmission of BFV.

### 2.2. DNA Extraction and Nested PCR

Genomic DNA was extracted from whole blood using a Wizard Genomic DNA Purification Kit (Promega, Fitchburg, MA, USA) or the magLEAD system (Precision System Science, Chiba, Japan) according to the manufacturer’s instructions. The DNA concentration was determined using a NanoDrop 8000 spectrophotometer (Thermo Fisher Scientific, Waltham, MA, USA), and samples were diluted to 20 ng/μL. The extracted DNA was screened for the *env* gene of BFV by nested PCR, and the primer sets used were described in a previous report [17].

### 2.3. Whole-Genome Amplification, Library Preparation and Next-Generation Sequencing

The nucleotide diversity among the six available full-genome sequences that we retrieved from GenBank was calculated with windows of 200 bases and a 25-base step using DnaSP v6.12 [26]. Several primers were designed from the most conserved region and both ends of BFV by using Primer3Plus software (http://www.bioinformatics.nl/cgi-bin/primer3plus/primer3plus.cgi (accessed on 14 May 2020)). The designed primer pairs were tested for PCR amplification using BFV-positive genomes, and the best pairs (BFV1-6061: Forward 5′-TGTGGTGGAAAGACCACCCGGAAATAAGCAAGGGC-3′, Reverse 5′-TGTCTTGTGTTGGGAGTGTTGTTCAGAGCAAG-3′; BFV5951-12001: Forward 5′-AAGCGGCTCTTAACGAAACTGTTGGC-3′, Reverse 5′-ATTGTTGTGACCTTCTCCAATCTTTAGTGGATTC-3′) were used for further studies. The amplification of two amplicons (BFV1-6052 and BFV5952-12001) covering whole viral genomes was performed in a reaction mixture containing 12.5 μL of PrimeSTAR Max Premix (TaKaRa Bio, Kusatsu, Japan), a primer pair (BFV1-6052 or BFV5952-12001) for each amplicon at 0.3 μM, 1 μL of the extracted DNA and enough PCR-grade water to reach a final volume of 25 μL. Amplification was performed under the following conditions: an initial PCR activation step at 94 °C for 5 s, 5 cycles of denaturation at 98 °C for 10 s and annealing and extension at 70 °C for 6 min, and 40 additional cycles with an annealing and extension temperature of 68 °C. Amplification of PCR products was confirmed by gel electrophoresis. The methods of purification of PCR products, library preparation and next-generation sequencing followed those in our previous report [27].

### 2.4. Data Analysis of Next-Generation Sequencing

The sequences generated by next-generation sequencing were analyzed using the CLC Genomics Workbench 11 software (Qiagen, Venlo, The Netherlands). The sequences were then processed to remove primers and low-quality sequences and mapped to reference genomes of the BFV strain NU (accession No. LC510607). The complete sequences of the BFVs that were determined in this study were submitted to the DDBJ under accession Nos. LC622406-35.

### 2.5. Phylogenetic Analysis

The BFV viral genome sequences that were obtained in this study and the available full-genome sequences that we retrieved from GenBank were aligned using ClustalW. Evolutionary distances were computed using Kimura’s two-parameter model. Molecular phylogenetic trees were constructed using the neighbor-joining method with MEGA7 software [28]. A total of 1000 bootstrap replicates were used to derive the phylogenetic trees.

### 2.6. Statistical Analysis

The chi-square test was used to compare the positive number of infected farms and cattle for each prefecture, age, and cattle breed. These analyses were performed using GraphPad Prism 6 software (GraphPad Software, San Diego, CA, USA). *p* < 0.05 was considered statistically significant in this study.

## 3. Results

In total, 12 of the 89 (13.4%) farms and 30 of the 884 (3.4%) cattle were positive for BFV according to nested PCR (Table 1). The number of positive farms and the positivity rate for each prefecture were as follows: Miyazaki (*n* = 8, 13.3%), Kumamoto (*n* = 1, 7.1%), Oita (*n* = 1, 100%), Okinawa (*n* = 0, 0%), Kagoshima (*n* = 0, 0%) and Hokkaido (*n* = 2, 16.6%). The number of positive cattle and the positivity rate for each prefecture were as follows: Miyazaki (*n* = 15, 2.5%), Kumamoto (*n* = 5, 3.5%), Oita (*n* = 7, 29.1%), Okinawa (*n* = 0, 0%), Kagoshima (*n* = 0, 0%), and Hokkaido (*n* = 3, 3.0%). The numbers of positive farms and cattle were compared among the Miyazaki, Kumamoto, and Hokkaido prefectures since only one farm was investigated in the Oita, Kagoshima, and Okinawa prefectures. No significant differences in the numbers of positive farms (*p* = 0.77) and cattle (*p* = 0.75) were confirmed among these prefectures. In this initial study, no PCR positivity was confirmed among cattle less than 48 months old (*n* = 255) (Table 2). Conversely, 4.9% (13/265), 4.3% (9/221), and 6.7% (7/110) of the cattle were PCR-positive at 48–84, 84–120, and more than 120 months old, respectively. A significant difference among the age groups (*p* = 0.0029) was observed, as was a higher rate of BFV infection in older animals. No difference in the BFV positivity rate was observed between Holstein (*n* = 4, 3.4%) and Japanese Black cattle (*n* = 25, 3.3%).

Previous studies reported that BFV was isolated from milk and uterine cells [14,29]. Although cattle less than 48 months old were scarcely confirmed to have BFV infection in this study, the virus was thought to have potential for dam-to-calf transmission. To investigate this possibility, we performed an additional survey in 77 pairs of dams and 3-month-old calves on a farm confirmed to have BFV infection. One of every seven calves born from a dam with BFV was infected with BFV. Thus, the possibility of dam-to-calf transmission of BFV was confirmed. BFVs were successfully isolated from three blood samples of seven PCR-positive dams. The three virus isolates were named BFV11996, BFV13294, and BFV13630 and were used for further experiments.

To design primers for amplifying the whole genome of BFV, we performed sliding window analysis among the six isolates of BFV with deposited whole-genome sequences in GenBank (accession Nos. AY134750, LC510606-7, NC001831, and JX307861-2) (Figure 2a). The sequences of the *pol* gene corresponding to positions 5887–6086 of the BFV Riems strain (accession No. JX307862) were the most conserved region among the six BFVs. Thus, we designed primers from this conserved region and amplified 32 pairs of BFV amplicons (Figure 2b). Finally, we obtained 30 complete consensus sequences for BFVs identified in the Miyazaki, Kumamoto, Oita, and Hokkaido prefectures in Japan.

The phylogenetic analysis of the whole viral sequence indicated that there were three genotypes among the BFVs (Figure 3). Most isolates in this study were of the Asia/USA genotype. Three isolates from Miyazaki and Hokkaido were of the European genotype. Furthermore, a novel genotype, named “Third genotype”, was identified, and four isolates from Oita were in this group. The nucleotide distances between isolates in the Europe and Third, Europe and Asia/USA, and Third and Asia/USA genotypes were in the ranges 2.9–3.3%, 3.6–4.2%, and 3.8–4.2%, respectively. The nucleotide similarity rates of LTR and *gag*, *pol*, *env*, *bel-1* and *bel-2* genes among all isolates were 88.6%, 91.1%, 90.6%, 89.6%, 92.4%, and 92.8%, respectively. The deletion of 12 nucleotides (amino acid position of Gag protein: 170–173) was confirmed in three isolates of BFV (BFV/Kumamoto/A-6/2019, BFV/Kumamoto/A-7/2019, and BFV/Miyazaki/J-5/2018). A nucleotide substitution of a potential initiator codon in the *bel-2* gene (corresponding to position 10,122 of the BSV Riems strain), resulting in protein substitution from methionine to isoleucine, was confirmed in three isolates of BFV of the European genotype (BFV/Miyazaki/I-9/2018, BFV/Miyazaki/D-9/2020, and BFV/Hokkaido/L-72/2016). 

Finally, we performed sliding window analysis among the 36 full-genome sequences of BFV (Figure 2c). The most variable region was confirmed in the LTR region corresponding to positions 1037–1236 of the BFV Riems strain. Conversely, the most conserved region was confirmed in the *pol* region corresponding to positions 5749–5948 of the BFV Riems strain.

## 4. Discussion

This is a report of a molecular epidemiological study of BFV in the Kyushu and Hokkaido areas in Japan. We revealed that 3.4% of cattle are infected with BFV. The whole viral sequences of 30 isolates were determined, and three genotypes, including a novel genotype, were confirmed to be endemic in Japan.

To reveal the prevalence of BFV in Japan, we investigated 784 samples from 77 farms in the Kyushu area (Miyazaki, Kagoshima, Kumamoto, Oita, and Okinawa) and 100 samples from 12 farms in the Hokkaido area (Table 1 and Figure 1). The positivity rates were similar in the Kyushu (3.4%; 27/784) and Hokkaido (3.0%; 3/100) areas. A previous study reported that 16.7% of cattle (91/545, 20 farms) in the Kanto area in Japan were infected with BFV [18]. This previous report included an increased number of collected samples if BFV-positive cattle were detected on a farm. Thus, the prevalence of BFV might have been slightly overestimated. Nevertheless, the rate of BFV infection could be different among areas in Japan due to the difference in stocking density, the average number of feeding years and the frequency of movement among farms. The prevalence rates of BFV in Germany (7%) and Poland (30–42%) were previously reported [14,15]. All the surveys conducted in Japan used PCR methods, and Iwasaki et al. confirmed that positive results were coincident when using agar gel immunodiffusion (AGID) and PCR tests [18]. The surveys conducted in foreign countries used a variety of serological methods, mainly ELISA tests. Generally, ELISA has a higher sensitivity than AGID tests. Thus, the epidemiological surveys conducted in foreign countries might have higher sensitivity than those conducted in Japan.

A significant difference was found between the age of cattle and BFV positivity in this study. A similar tendency was reported in previous studies [16,19]. This tendency suggests that BFV is mainly transmitted through the horizontal route under certain conditions. In contrast, some studies conducted in other countries reported no difference between the age of cattle and BFV positivity [14]. This result might be affected by the breeding style of each country, especially the duration of the feeding period with dams and calves. Simian and feline FVs are released from the oral mucosa and thought to be transmitted through biting [11,12,30]. However, cows are milder animals and rarely attack by biting. A previous study confirmed that spraying the infectious agent into the throat was the most successful route of BFV infection [13]. Thus, BFV might be horizontally transmitted by licking and sneezing. Cattle less than 48 months old were scarcely PCR-positive in this study. On the other hand, BFV is thought to have the potential for vertical transmission due to virus isolation from milk and the placenta and uterus [14,29]. Thus, BFV infection of 77 pairs of dams and calves from a farm with BFV infection was confirmed. We used 3-month-old calves because vertical transmission of *Theileria orientalis* was not detectable in 1-month-old calves, but was detectable in 3-month-old calves by PCR [31]. One dam and calf pair was infected with BFV. Thus, BFV might be transmitted via a vertical route, but the frequency is considered to be very low considering the prevalence among young cattle.

BFV was divided into two genotypes (Europe and Asia/USA) in the past [19,32]. In addition, this study identified a novel genotype, named the Third genotype (Figure 3). The phylogenetic analyses based on the sequences of all the coding and noncoding regions of BFV also classified all the isolates into identical genotypes based on the whole viral sequence (Figure 4). Thus, the Third genotype is thought to be completely branched and established in BFV. Several serotypes have been reported in simian and and two in feline FV [33,34]. The nucleotide distance between the two serotypes of feline FV was approximately 7%. The nucleotide distances among the three genotypes of BFV was in the range 2.9–4.2%. Thus, we assume that there is no serological difference among the BFV genotypes. However, further studies are required. Although we identified four isolates of BFV clustered into the Third genotype, all isolates were identified on the same farm in Oita prefecture. Thus, we could not conclude that this genotype was spread throughout Japan. The Asia/USA genotype can be divided into two clusters based on whole viral sequences (Figure 3). The phylogenetic trees based on the sequences of the LTR and *gag*, *pol,* and *env* genes can also be divided into two clusters (Figure 4a–d). However, the phylogenetic trees based on the sequences of the *bel-1* and *bel-2* genes could not be divided because these genes were the most conserved regions (Figure 4e,f). Thus, branching of the two clusters might have occurred in recent decades and may be currently in progress. A similar tendency was observed in the phylogenetic analysis of bovine coronavirus [35]. Many Holstein cattle were imported from North America to Japan more than 20 years ago. However, few live cattle have been imported into Japan, except for importation which has occured from Australia in the last two decades due to the problems of transboundary diseases, such as foot-and-mouse disease and bovine spongiform encephalopathy. This policy might affect the branching of the Asia/USA genotype in Japan. 

The deletion of four amino acids in the Gag protein (170–173) was confirmed in three isolates of BFV. Amino acid exchange in the BFV Gag protein was reported to correlate with efficient amplification and cell-free transmission of BFV in vitro [36,37]. Although the deletion of four amino acids was not observed in previous studies, it might affect the amplification and transmission of BFV. The nucleotide similarities of *env* (89.6%) and *bel-2* (92.8%) were the lowest and highest among BFVs. Although *bel-2* was the most conserved gene, protein substitution of a potential initiator codon of Bel-2 protein was confirmed among three isolates of the European genotype identified in Japan. Bel-2 protein has not been unambiguously identified in simian, feline and bovine FVs. Thus, this observation supports the idea that the bel-2 protein does not exist in FVs. Conversely, the Bet protein is encoded by a spliced mRNA from the first part of the sequence of *bel-1* and the entire sequence of *bel-*2 [4]. The amino substituted acid from methionine to isoleucine was located downstream of the splice acceptor [38]. Thus, the mutation of the potential initiator codon in the *bel-2* gene would not affect Bet protein expression.

## 5. Conclusions

We found that the prevalence of BFV in Japan is 3.4%, and three genotypes (Asia/USA, Europe, and Third), including a novel genotype (Third), are spread in Japan. To date, there is not enough information available regarding this virus throughout the world. Genetic information about BFV is important for the understanding viral evolution and the possibility of its effective utilization as a novel vector. Therefore, further studies are required to identify the importance and evolution of this virus.

## Figures and Tables

**Figure 1 viruses-13-01017-f001:**
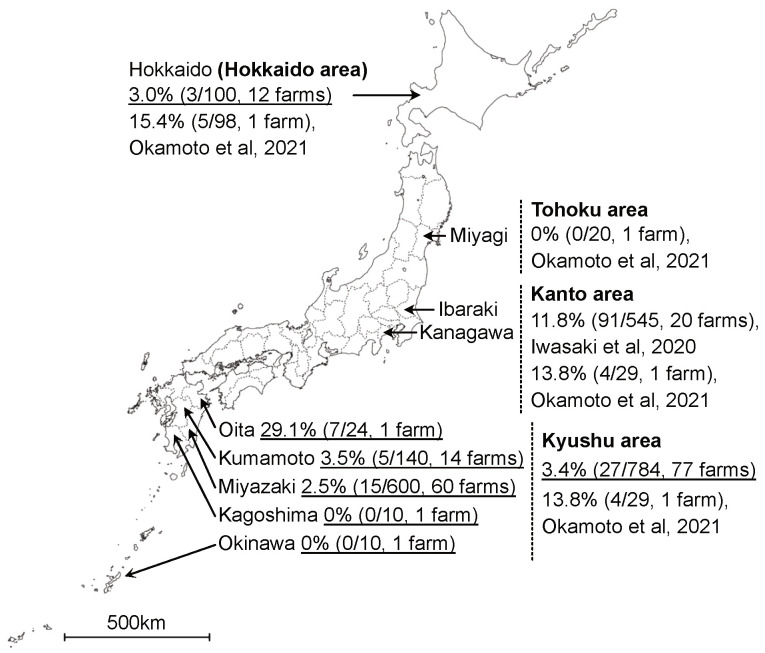
Map of Japan showing the positivity rate of bovine foamy virus (BFV) for each area. The map indicates the six prefectures (Hokkaido, Oita, Kumamoto, Miyazaki, Kagoshima and Okinawa) where sampling was performed in this study, and three prefectures (Miyagi, Ibaraki and Kanagawa) where sampling was performed in previous studies [18,19]. The BFV positivity rate is indicated by %, and the parentheses show the numbers of positive and tested samples and the number of tested farms in each prefecture and area. The surveys conducted by this study are indicated with underlines.

**Figure 2 viruses-13-01017-f002:**
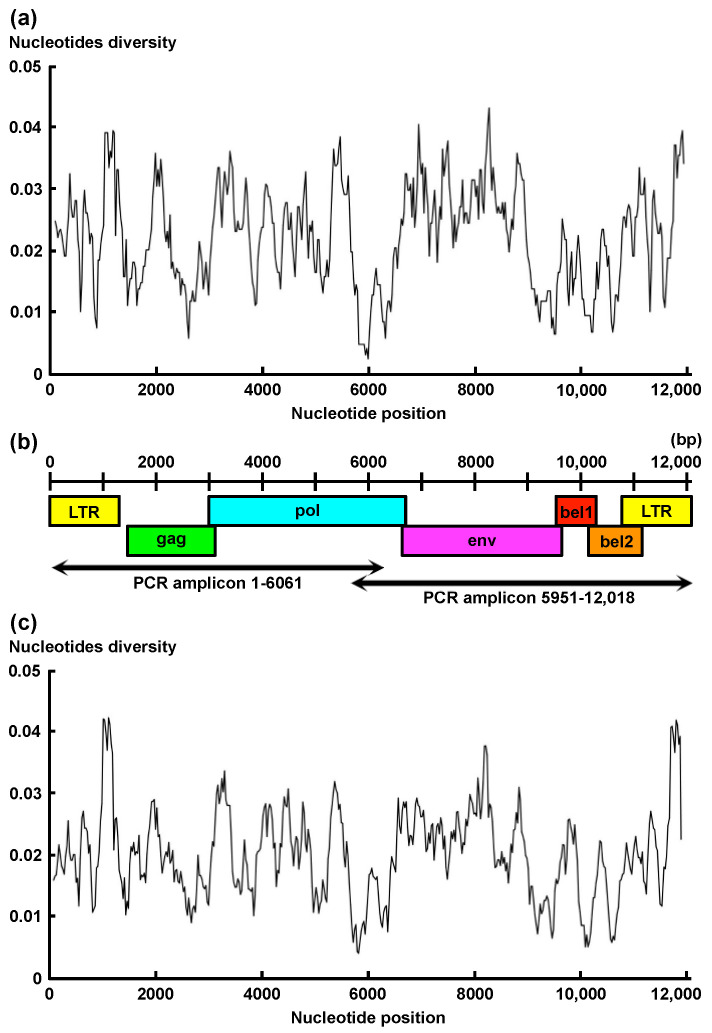
Genome structure and nucleotide diversity of bovine foamy virus (BFV). The nucleotide diversity among the six full-genome sequences of BFVs deposited in GenBank (accession Nos. AY134750, LC510606-7, NC001831, and JX307861-2) is presented. The nucleotide diversity was calculated using DnaSP v6.12 [26]. A sliding window method with windows of 200 nucleotides and steps of 25 nucleotides was used to calculate the nucleotide diversity across the entire alignment (**a**). Genome structure and locations of PCR amplicons for whole-genome analysis are presented. The BFV genome consists of approximately 12,000 nucleotides, including three structural genes (*gag*, *pol*, *env*) and two accessory genes (*bel-1*, *bel-2*) flanked by two identical long terminal repeats (LTRs). The locations of the two PCR amplicons for whole-genome analysis are shown (**b**). The nucleotide diversity of full-genome sequences among the 36 BFVs that were obtained in this study and deposited in GenBank is presented (**c**).

**Figure 3 viruses-13-01017-f003:**
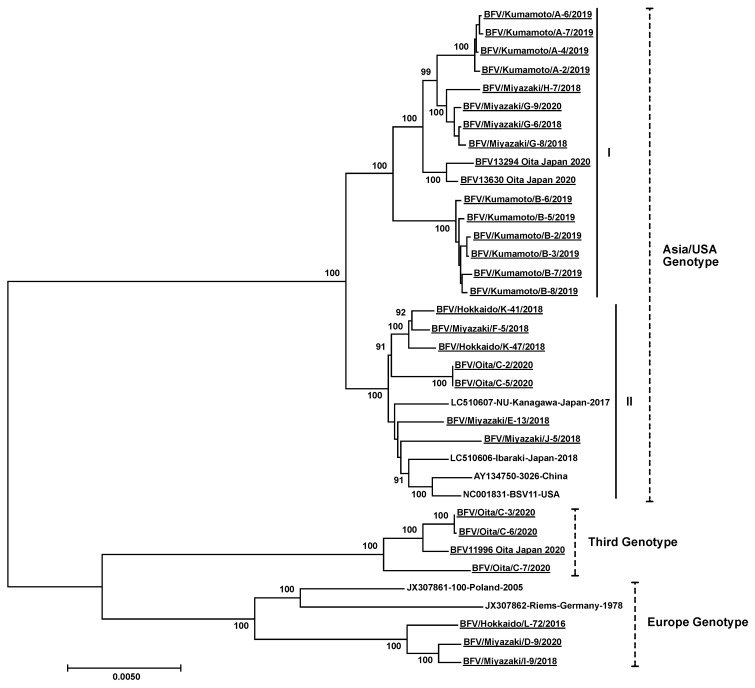
Phylogenetic analysis of the whole-genome sequences of bovine foamy virus (BFV). The analysis involved the whole-genome sequences of the 36 BFV isolates at the nucleotide level. The names of the 27 isolates for which the whole viral sequence was identified in this study are denoted as “BFV”, “isolation prefecture”, “name of isolate”, and “isolation year” separated by slashes and with underlines (e.g., BFV/Miyazaki/G-6/2018). Three isolates of the virus isolated in this study are named as “name of isolate”, “isolation prefecture”, “Japan”, and “isolation year” with underlines (e.g., BFV13294 Oita Japan 2020). The sequences retrieved from GenBank are denoted as “accession No.”, “name of strain”, “isolation (prefecture) country”, and “isolation year” separated by hyphens (e.g., LC150607-NU-Kanagawa-Japan-2017). Bootstrap values of more than 80% (1000 replicates) are shown next to the branches. The scale bar indicates the number of substitutions per site. The vertical broken and full bars show the classification of genotypes (Asia/USA, Third, and Europe) and clusters (I, II) of BFV, respectively.

**Figure 4 viruses-13-01017-f004:**
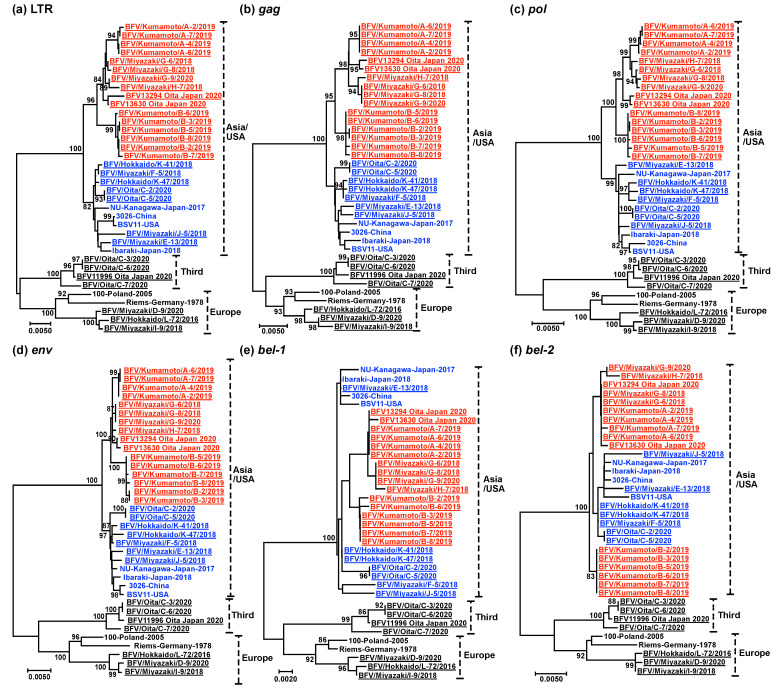
Phylogenetic analyses of the entire coding and noncoding genes of bovine foamy virus (BFV). Phylogenetic trees based on the entire LTR (**a**) and *gag* (**b**), *pol* (**c**), *env* (**d**), *bel-1* (**e**), and *bel-2* (**f**) genes. The 27 isolates for which the whole viral sequence was identified in this study are denoted as “BFV”, “isolation prefecture”, “name of isolate”, and “isolation year” separated by slashes and with underlines (e.g., BFV/Miyazaki/G-6/2018). Three isolates of the virus isolated in this study are denoted as “name of isolate”, “isolation prefecture”, “Japan”, and “isolation year” with underlines (e.g., BFV13294 Oita Japan 2020). The sequences retrieved from GenBank are denoted as “name of isolate”, “isolation (prefecture) country”, and “isolation year” with hyphens (e.g., NU-Kanagawa-Japan-2017). Bootstrap values of more than 80% (1000 replicates) are shown next to the branches. The scale bars indicate the number of substitutions per site. The vertical broken bars show the classification of genotypes (Asia/USA, Third, and Europe). The clusters (I, II) of BFV within the Asia/USA genotype are colored with red and blue font.

**Table 1 viruses-13-01017-t001:** Prevalence of bovine foamy virus on 89 farms in the Hokkaido and Kyushu areas in Japan.

Prefecture	Area	No. of Positive Farms/Tests (Positive Rate)	No. of Positive Cattle/Tests (Positive Rate)
Miyazaki	Kyushu	8/60 (13.3%)	15/600 (2.5%)
Kumamoto	Kyushu	1/14 (7.1%)	5/140 (3.5%)
Oita	Kyushu	1/1 (100%)	7/24 (29.1%)
Okinawa	Kyushu	0/1 (0%)	0/10 (0%)
Kagoshima	Kyushu	0/1 (0%)	0/10 (0%)
Hokkaido	Hokkaido	2/12 (16.6%)	3/100 (3.0%)
Total		12/89 (13.4%)	30/884 (3.4%)

**Table 2 viruses-13-01017-t002:** Prevalence of bovine foamy virus among the age groups of cattle.

Age (Months)	No. of Samples	No. of Positive Cattle (%)
0–48	255	0 (0)
48–84	265	13 (4.9)
84–120	221	9 (4.3)
>120	110	7 (6.7)
Unknown	33	1 (3.0)
Total	884	30 (3.4)

## Data Availability

The data that support the findings of this study are available from the corresponding author, H.M., upon reasonable request. The whole viral sequences of BFV obtained in this study are deposited in DDBJ (accession Nos. LC622406-35) and available at https://www.ddbj.nig.ac.jp/index-e.html (accessed on 28 May 2021).

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
