# Peer review of "Molecular Epidemiology and Whole-Genome Analysis of Bovine Foamy Virus in Japan"

_viruses, 2021, doi:10.3390/v13061017_

Round 1

Reviewer 1 Report

The manuscript "Molecular Epidemiology and Whole-Genome Analysis of Bovine Foamy Virus in Japan" by Mekata et al. provides new data on the epidemiology and genetic, molecular and evolutionary features of bovine foamy virus (BFV) isolates from Japan.

The manuscript contains a lot of sequencing data and bioinformatics on the newly gained BFV sequences from different parts of Japan. The authors interpret their data mainly with respect to BFV epidemiology but leave out several other interesting aspects related to phylogeny and biology of BFVs. Importantly, they ignore sequencing data from other Japanese BFV isolates, these sequences have to be included in order to improve the value of the data presented and the conclusions drawn.

In general, the data are convincing, data presentation can be improved as suggested below and a thorough language check seem absolutely necessary.

Major / general points of concerns:

  1. The authors have to include the additional sequencing data from Japanese BFV isolates as described by Okamoto et al., 2020 and Hachiya et al., 2018 into their analyses in order to obtain a much more comprehensive image and to avoid the impression that related studies had NOT been conducted by others before. In particular, the previously described sequences data have to be included into all phylogenetic analyses in order to extend the prior findings by Okamoto et al., Fig. 3, which is very similar to the data described here in Figs. 2 and 3. These prior data have to be also properly cited and acknowledged here.
  2. A map depicting the sampling sites in this and the previous studies should be provided in order to conduct studies on geographic clustering and/or isolation of the different BFV variants defined here and in previous studies.
  3. The data presentation (improvement of the text and inclusion of a suited figure) concerning the apparent loss of a bel2 initiator codon has to be improved. The authors have to properly indicate whether this start codon in question is up- or downstream of the bet splice acceptor in bel2. In addition, they have to mention that a Bel 2 protein initiated at this or other bel2 Met codons has not been described yet in any FV known – thus, this whole point may be irrelevant concerning BFV bel protein expression.
  4. The authors provide the impression that the previous epidemiological studies from Japan have been very small compared to theirs – this is not true (884 animals here compared to 181 plus 545 in previous studies). In addition, a critical evaluation on the differences to the previous Japanese and international studies on BFV epidemiology is mostly lacking, for instance concerning methodology (ELISA vs. PCR etc), animal husbandry, sampling etc.
  5. The authors have to carefully use the terms isolate, variant, genotype etc. To the understanding of the reviewer and in line with the current literature, the authors obtained and characterized different “BFV isolates” which belong to different “BFV clusters or clades”. Please use the proper terms consistently throughout the whole manuscript (e.g. lines 19, 64, 197ff, 248, 282ff, etc.)
  6. The authors should (have to) sub-divide the text into smaller paragraphs in order to structure data presentation and discussion.

Specific comments:

  1. Line 20 and elsewhere: This statement (especially in the abstract) is wrong since the authors report about a calf that was BFV-positive by PCR. Omit this statement in the abstract and correct throughout the whole manuscript.
  2. Line 53-55: Please correct this statement as mentioned above.
  3. Line 139: In this initial study, ………………
  4. Line 152 ff.: Wouldn’t it be more meaningful to express that approx. 14% of the BFV-positive dams had a BFV –positive calve, the overall percentage of BFV–positive calves is not very informative. The authors mention in the discussion that in most cases, dams and calves are separated very early. Here, this was obviously not the case – so the authors should provide more information on whether and when dams and calves are separated
  5. Line 167: Replace: Most instead of Many?
  6. Line 169: please clarify like: ….., and all four were from a the only farm from the Oita prefecture. Is this a region/farm with special husbandry and trade connections/traditions?
  7. Line 174 ff: How does the Gag deletion relate to those of the German and Chinese high-titer variants selected for efficient cell-free transmission?
  8. Line 176 ff: Please, describe understandably where the methionine exchange in bel2 is located and whether it is up- or downstream of the bet splice acceptor. Here or in the discussion: please mention that a Bel2 protein has never been unambiguously detected in any of the FVs isolated in more detail – so, the importance of this finding is/may be disputable.
  9. 1: the Legend is displace from the figure by intervening text – please correct.
  10. Lines 212ff: please specify that this statement may be only true under the Japanese husbandry conditions.
  11. Lines 214: again, a map with the sampling sites would be desirable.
  12. Line 218 ff: Please improve this sentence, it is hard to get what you mean.
  13. Line 226: …… through the horizontal route under certain conditions.
  14. Line 234: as above – this sentence is wrong since, overall, you detected at least one BFV-positive calf. Please adapt also the following sentences.
  15. Lines 250-256: The text is not really clear here – please specify in more detail and length – the current text is complicated to understand.
  16. Lines 256 ff: please provide statistical evidence that bel1/bel2 sequences of the Asian/American cluster have in fact a different evolutionary history than the rest of the genome. Please, conduct breakpoint analyses to address your point. In addition, please include in all these studies also the other published BFV sequences from Japan in order to see whether there is in fact a clear trend.
  17. Lines 262ff and 251ff: please bring both statements on the impact of cattle import on Japanese cattle phylogeny into a single and easy to grasp statement and only then relate this to your finding covering all available BFV sequences from Japan and elsewhere.
  18. Lines 268ff: bring the discussion on bel2/bet into a separate paragraph, include a figure to depict your findings and describe/discuss that a Bel2 protein has not been unambiguously identified in PFV/SFVs and FFV, the best studied FVs in this respect. Assuming that BFV Bet has more or less the same expression strategy (via splicing), structure (see e.g. Slavkovic-Lukic et al., 2013) and function as PFV/SFV/FFV Bet, the mutation of a potential initiator codon cannot affect overall Bet expression as long as splicing is not affected by these changes. The authors should re-consider their ideas under these aspects.
  19. 3: Some sort of color-coding of the different sub-clusters within the Asian-America BFV groups would be helpful to follow your data interpretation.

Reviewer 2 Report

Viruses-1213727-Review

Mekata et al. performed a seroprevalence survey of bovine foamy virus infection in Japanese cattle. They report modest infection rates, and higher prevalence in older animals. Full length sequence of 30 viruses were reported with phylogenetic analyses. The work is well performed, data are clearly presented and appropriately discussed. The manuscript could be improved with use of more accurate vocabulary, avoidance of overstatements, and updating the bibliography.

Line 26: the conclusions on transmission are overstatement. The authors described prevalence in different age groups that support horizontal transmission (cohort one) and one calf which has probably be infected by the vertical route (cohort 2).

Lines 31-48 : data on miRNA encoded by BFV should be cited (A. W. Whisnant, T. Kehl, Q. Y. Bao, M. Materniak, J. Kuzmak, M. Lochelt, et al.  Journal of Virology 2014 Vol. 88 Issue 9 Pages 4679-4686 and following papers from the same group).

Line 53 “wide range of area” Line 54 “ the limited number of area”. The two sentences are contradictory. It will help the reader to have quantitative information: number of regions in Japan, number in which epidemiological surveys have been conducted; range of prevalence observed.

Reference 17 is insufficient to describe the evolutionary history of foamy viruses. As one goal of the work is to document the evolution history of BFV, the scientific background should be more documented with up to date references on SFV and endogenous FV.

Line 65 “the effective utilization of BFV” is an overstatement. To the best of my knowledge, BFV based vectors have not yet been constructed and tested in vitro.

Two genotypes differing in env gene have been described in simian and feline foamy viruses (L. Richard, R. Rua, E. Betsem, A. Mouinga-Ondeme, M. Kazanji, E. Leroy, et al. Journal of Virology 2015 Vol. 89 Issue 24 Pages 12480-12491 - L. Richard, R. Rua, E. Betsem, A. Mouinga-Ondeme, M. Kazanji, E. Leroy, et al.Journal of Virology 2015 Vol. 89 Issue 24 Pages 12480-12491). These genotypes are important because they are targeted by neutralizing antibodies (C. Lambert, M. Couteaudier, J. Gouzil, L. Richard, T. Montange, E. Betsem, et al. PLoS Pathog 2018 Vol. 14 Issue 10 Pages e1007293; M. Zemba, A. Alke, J. Bodem, I. G. Winkler, R. L. P. Flower, K. I. Pfrepper, et al. 2000 Vol. 266 Issue 1 Pages 150-156). The authors should analyze whether two similar genotypes are present among BFV strains. Results should be discussed.

Minor comments

Line 22 “farm with BFV” sounds awkward.

Line 22 the dam and calf are paired, not their infection.

Line 15 Sentence on FV as viral vectors is out of context.  

Line 19 and line 22 The total number of animal tested should be indicated.

Round 2

Reviewer 1 Report

The manuscript "Molecular Epidemiology and Whole-Genome Analysis of Bovine Foamy Virus in Japan" by Mekata et al. provides new data on the epidemiology and genetic, molecular and evolutionary features of bovine foamy virus (BFV) isolates from Japan.

The manuscript contains a lot of sequencing data and bioinformatics on newly gained BFV sequences from different parts of Japan. The authors have invested some effort into improving the manuscript and followed most of the reviewers’ suggestions – thanks for this, however there is still a lot to do.

It appears to me (and I have to admit that I am not a native-speaker) that language is still a big issue. To my understanding, proper phrasing and understandability / clarity is unfortunately still the key issue – we are not talking about language polishing. Since your professional support did not help too much, I would suggest that the authors seek help by the publisher and the support provided by the journal. I understand that one does not like to invest additional expenses into this but it appears to be necessary here.

General and specific points of concerns:

  1. I would suggest to shift the map with the sampling sites into the introduction where these issues are presented for the previous and current studies (line 57 ff.).
  2. Line 34: …… (FV) sub-family ………….
  3. Line 40: The bel2 gene may encode the Bel2 protein but such a protein has never unambiguously detected in any FV analysed so far (REFERENCES).
  4. Line 59: ….. prevalence ranged ……..
  5. Lines 76ff, 137ff etc. : refer to the map
  6. Line 161: …… with BFV was infected with BFV.
  7. Line 163: ……….. of BFV was confirmed.
  8. Line 167: isolates instead of strains.
  9. Line 184: …… substitution of a potential initiator ………..
  10. Line 187: …... of BFV of the European …..
  11. 1b: typo: amplicon (twice below the genome in the figure).
  12. Lines 233-236: Please, comment on whether there are differences between both diagnostic methods, e.g. sensitivity etc.
  13. Sentences line 240ff: please clarify – I cannot follow your discussion / interpretation. Overall, the whole paragraph needs restructuring and clarity.
  14. Paragraph line 258ff: I dare to question whether there is in fact a ‘Third Genotype’. In my eyes, the ‘Third Genotype’ is only a comparably early branching off from the European clade. Taking together the old and revised texts related to the import of cattle into Japan, I understood that there had been an early import of cattle from Europe and a more recent (and more numerous) import from the US. If so (and please clarify this point), could this have led to the following scenario: an early import of cattle with European clade BFV led to a comparably deep-rooted branching off from the European clade (with low degree of admixture and only with few herds affected). This was followed by a much later (within the last 20 years as you mention) and bigger/numerous import of cattle from the US with animals and BFV variants that are different but more similar to the Japanese cattle since both belong to the American – Asian clade. PLEASE CLARIFY since there is a lack of clarity in your text and your argumentation.

In addition, please clarify the issue concerning the FFV and SFV serotypes: this is only due to a polymorphism of a defined segment of the SU ecto-domain and thus your discussion of overall similarity does not fit here.

  1. 3: sometimes, different names for the BFV isolates are used in the different panels – pls. correct. In addition: in the American-Asian clade: there is one name in black – what does this mean, this sequence is not present in the other trees – clarify and carefully correct.

In addition, the members in the red and blue groups are always the same for the different trees or does colour-coding differ between panels? All this is not clear and data presentation is not helpful – use the legend to clarify.

Please correct the whole figure really carefully and adapt your text/statements correspondingly. Based on your data presentation, I would assume that all members of this American/Asian BFV group show an identical sub-clustering for all genes/sequences analysed – is this so?

  1. Paragraph 288ff: Please indicate whether the deletions in gag are in that region of Gag that showed adaptive changes in the different high-titer BFV variants - is this so and if so, please indicate.

Please clarify also the discussion concerning the bel2 M/I exchanges and the potential relevance and whether this finding is really worth to discuss here.

  1. Please improve the final, summarizing sentences of the discussion (and abstract/introduction) to summarize there only the real big points.
